# Private Isotonic Regression

**Badih Ghazi**          **Pritish Kamath**          **Ravi Kumar**          **Pasin Manurangsi**

Google Research
Mountain View, CA, US
badihghazi@gmail.com, pritish@alum.mit.edu,
ravi.k53@gmail.com, pasin@google.com

## Abstract

In this paper, we consider the problem of differentially private (DP) algorithms for isotonic regression. For the most general problem of isotonic regression over a partially ordered set (poset) $\mathcal{X}$ and for any Lipschitz loss function, we obtain a pure-DP algorithm that, given $n$ input points, has an expected excess empirical risk of roughly $\mathrm{width}(\mathcal{X}) \cdot \log |\mathcal{X}|/n$, where $\mathrm{width}(\mathcal{X})$ is the width of the poset. In contrast, we also obtain a near-matching lower bound of roughly $(\mathrm{width}(\mathcal{X}) + \log |\mathcal{X}|)/n$, that holds even for approximate-DP algorithms. Moreover, we show that the above bounds are essentially the best that can be obtained without utilizing any further structure of the poset. In the special case of a totally ordered set and for $\ell_1$ and $\ell_2^2$ losses, our algorithm can be implemented in near-linear running time; we also provide extensions of this algorithm to the problem of private isotonic regression with additional structural constraints on the output function.

## 1   Introduction

Isotonic regression is a basic primitive in statistics and machine learning, which has been studied at least since the 1950s [4, 9]; see also the textbooks on the topic [5, 38]. It has seen applications in numerous fields including medicine [31, 39] where the expression of an antigen is modeled as a monotone function of the DNA index and WBC count, and education [19], where isotonic regression was used to predict college GPA using high school GPA and standardized test scores. Isotonic regression is also arguably the most common non-parametric method for calibrating machine learning models [51], including modern neural networks [23].

In this paper, we study isotonic regression with a differential privacy (DP) constraint on the output model. DP [17, 16] is a highly popular notion of privacy for algorithms and machine learning primitives, and has seen increased adoption due to its powerful guarantees and properties [37, 43]. Despite the plethora of work on DP statistics and machine learning (see Section 5 for related work), ours is, to the best of our knowledge, the first to study DP isotonic regression.

In fact, we consider the most general version of the isotonic regression problem. We first set up some notation to describe our results. Let $(\mathcal{X}, \leq)$ be any partially ordered set (*poset*). A function $f : \mathcal{X} \to [0, 1]$ is *monotone* if and only if $f(x) \leq f(x')$ for all $x \leq x'$. For brevity, we use $\mathcal{F}(\mathcal{X}, \mathcal{Y})$ to denote the set of all monotone functions from $\mathcal{X}$ to $\mathcal{Y}$; throughout, we consider $\mathcal{Y} \subseteq [0, 1]$.

Let $[n]$ denote $\{1, \ldots, n\}$. Given an input dataset $D = \{(x_1, y_1), \ldots, (x_n, y_n)\} \in (\mathcal{X} \times [0, 1])^n$, let the *empirical risk* of a function $f : \mathcal{X} \to [0, 1]$ be $\mathcal{L}(f; D) := \frac{1}{n} \sum_{i \in [n]} \ell(f(x_i), y_i)$, where $\ell : [0, 1] \times [0, 1] \to \mathbb{R}$ is a *loss function*.

We study private isotonic regression in the basic machine learning framework of empirical risk minimization. Specifically, the goal of the *isotonic regression* problem, given dataset $D = \{(x_1, y_1), \ldots, (x_n, y_n)\} \in (\mathcal{X} \times [0, 1])^n$, is to find a monotone function $f : \mathcal{X} \to [0, 1]$ that

36th Conference on Neural Information Processing Systems (NeurIPS 2022).

minimizes $\mathcal{L}(f; D)$. The *excess empirical risk* of a function $f$ is defined as $\mathcal{L}(f; D) - \mathcal{L}(f^*; D)$ where $f^* := \operatorname{argmin}_{g \in \mathcal{F}(\mathcal{X}, \mathcal{Y})} \mathcal{L}(g; D)$.

## 1.1 Our Results

**General Posets.** Our first contribution is to give nearly tight upper and lower bounds for any poset, based on its *width*, as stated below (see Section 4 for a formal definition.)

**Theorem 1** (Upper Bound for General Poset). *Let $\mathcal{X}$ be any finite poset and let $\ell$ be an $L$-Lipschitz loss function. For any $\varepsilon \in (0, 1]$, there is an $\varepsilon$-DP algorithm for isotonic regression for $\ell$ with expected excess empirical risk at most* $O\left( \frac{L \cdot \text{width}(\mathcal{X}) \cdot \log |\mathcal{X}| \cdot (1 + \log^2(\varepsilon n))}{\varepsilon n} \right)$.

**Theorem 2** (Lower Bound for General Poset; Informal). *For any $\varepsilon \in (0, 1]$ and any $\delta < 0.01 \cdot \varepsilon / |\mathcal{X}|$, any $(\varepsilon, \delta)$-DP algorithm for isotonic regression for a "nice" loss function $\ell$ must have expected excess empirical risk* $\Omega\left( \frac{\text{width}(\mathcal{X}) + \log |\mathcal{X}|}{\varepsilon n} \right)$.

While our upper and lower bounds do not exactly match because of the multiplication-vs-addition of $\log |\mathcal{X}|$, we show in Section 4.3 that there are posets for which each bound in tight. In other words, this gap cannot be closed for generic posets.

**Totally Ordered Sets.** The above upper and lower bounds immediately translate to the case of totally ordered sets, by plugging in $\text{width}(\mathcal{X}) = 1$. More importantly, we give efficient algorithms in this case, which runs in time $\tilde{O}(n^2 + n \log |\mathcal{X}|)$ for general loss function $\ell$, and in nearly linear $\tilde{O}(n \cdot \log |\mathcal{X}|)$ time for the widely-studied $\ell_2^2$- and $\ell_1$-losses[1].

**Theorem 3.** *For all finite totally ordered sets $\mathcal{X}$, $L$-Lipschitz loss functions $\ell$, and $\varepsilon \in (0, 1]$, there is an $\varepsilon$-DP algorithm for isotonic regression for $\ell$ with expected excess empirical risk $O\left( \frac{L \cdot (\log |\mathcal{X}|) \cdot (1 + \log^2(\varepsilon n))}{\varepsilon n} \right)$. The running time of this algorithm is $\tilde{O}(n^2 + n \log |\mathcal{X}|)$ in general and can be improved to $\tilde{O}(n \log |\mathcal{X}|)$ for $\ell_1$ and $\ell_2^2$ losses.*

We are not aware of any prior work on private isotonic regression. A simple baseline algorithm for this problem would be to use the exponential mechanism over the set of all monotone functions taking values in a discretized set, to choose one with small loss. We show in Appendix A that this achieves an excess empirical risk of $O(L \cdot \sqrt{\text{width}(\mathcal{X}) \cdot \log |\mathcal{X}| / \varepsilon n})$, which is quadratically worse than the bound in Theorem 1. Moreover, even in the case of a totally ordered set, it is unclear how to implement such a mechanism efficiently.

We demonstrate the flexibility of our techniques by showing that it can be extended to variants of isotonic regression where, in addition to monotonicity, we also require $f$ to satisfy additional properties. For example, we may want $f$ to be $L_f$-Lipchitz for some specified $L_f > 0$. Other constraints we can handle include $k$-piecewise constant, $k$-piecewise linear, convexity, and concavity. For each of these constraints, we devise an algorithm that yields essentially the same error compared to the unconstrained case and still runs in polynomial time.

**Theorem 4.** *For all finite totally ordered sets $\mathcal{X}$, $L$-Lipschitz loss functions $\ell$, and $\varepsilon \in (0, 1]$, there is an $\varepsilon$-DP algorithm for $k$-piecewise constant, $k$-piecewise linear, Lipchitz, convex, or concave isotonic regression for $\ell$ with expected excess empirical risk $\tilde{O}\left( \frac{L \cdot (\log |\mathcal{X}|)}{\varepsilon n} \right)$. The running time of this algorithm is $(n |\mathcal{X}|)^{O(1)}$.*

**Organization.** We next provide necessary background on DP. In Section 3, we prove our results for totally ordered sets (including Theorem 3). We then move on to discuss general posets in Section 4. Section 5 contains additional related work. Finally, we conclude with some discussion in Section 6. Due to space constraints, we omit some proofs from the main body; these can be found in the Appendix.

---

[1]Recall that the $\ell_2^2$-loss is $\ell_2^2(y, y') = (y - y')^2$ and the $\ell_1$-loss is $\ell_1(y, y') = |y - y'|$.

## 2 Background on Differential Privacy

Two datasets $D = \{((x_1, y_1), \ldots, (x_n, y_n)\}$ and $D' = \{(x'_1, y'_1), \ldots, (x'_n, y'_n)\}$ are said to be *neighboring*, denoted $D \sim D'$, if there is an index $i \in [n]$ such that $(x_j, y_j) = (x'_j, y'_j)$ for all $j \in [n] \setminus \{i\}$. We recall the formal definition of differential privacy [18, 16]:

**Definition 5** (Differential Privacy (DP) [18, 16]). *Let $\varepsilon > 0$ and $\delta \in [0, 1]$. A randomized algorithm $\mathcal{M} : \mathcal{X}^n \to \mathcal{Y}$ is $(\varepsilon, \delta)$-differentially private ($(\varepsilon, \delta)$-DP) if, for all $D \sim D'$ and all (measurable) outcomes $S \subseteq \mathcal{Y}$, we have that $\Pr[\mathcal{M}(D) \in S] \leq e^{\varepsilon} \cdot \Pr[\mathcal{M}(D') \in S] + \delta$.*

We denote $(\varepsilon, 0)$-DP as $\varepsilon$-DP (aka *pure*-DP). The case when $\delta > 0$ is referred to as *approximate*-DP.

We will use the following composition theorems throughout our proofs.

**Lemma 6.** *$(\varepsilon, \delta)$-DP satisfies the following:*

- **Basic Composition:** *If mechanisms $\mathcal{M}_1, \ldots, \mathcal{M}_t$ are such that $\mathcal{M}_i$ satisfies $(\varepsilon_i, \delta_i)$-DP, then the composed mechanism $(\mathcal{M}_1(D), \ldots, \mathcal{M}_t(D))$ satisfies $(\sum_i \varepsilon_i, \sum_i \delta_i)$-DP. This holds even under adaptive composition, where each $\mathcal{M}_i$ can depend on the outputs of $\mathcal{M}_1, \ldots, \mathcal{M}_{i-1}$.*
- **Parallel Composition [33]:** *If a mechanism $\mathcal{M}$ satisfies $(\varepsilon, \delta)$-DP, then for any partition of $D = D_1 \sqcup \cdots \sqcup D_t$, the composed mechanism given as $(\mathcal{M}(D_1), \ldots, \mathcal{M}(D_t))$ satisfies $(\varepsilon, \delta)$-DP.*

**Exponential Mechanism.** The exponential mechanism solves the basic task of *selection* in data analysis: given a dataset $D \in \mathcal{Z}^n$ and a set $\mathcal{A}$ of options, it outputs the (approximately) best option, where "best" is defined by a *scoring function* $\mathfrak{s} : \mathcal{A} \times \mathcal{Z}^n \to \mathbb{R}$. The $\varepsilon$-DP *exponential mechanism* [34] is the randomized mechanism $\mathcal{M} : \mathcal{Z}^n \to \mathcal{A}$ given by

$$\forall D \in \mathcal{Z}^n, a \in \mathcal{A} : \Pr[\mathcal{M}(D) = a] \propto \exp\left(-\tfrac{\varepsilon}{2\Delta_{\mathfrak{s}}} \cdot \mathfrak{s}(a, D)\right),$$

where $\Delta_{\mathfrak{s}} := \sup_{D \sim D'} \max_{a \in \mathcal{A}} |\mathfrak{s}(a, D) - \mathfrak{s}(a, D')|$ is the *sensitivity* of the scoring function.

**Lemma 7** ([34]). *For $\mathcal{M}$ being the $\varepsilon$-DP exponential mechanism, it holds for all $D \in \mathcal{Z}^n$ that*

$$\mathbb{E}[\mathfrak{s}(\mathcal{M}(D), D)] \leq \min_{a \in \mathcal{A}} \mathfrak{s}(a, D) + \tfrac{2\Delta_{\mathfrak{s}}}{\varepsilon} \log |\mathcal{A}|.$$

**Lower Bound for Privatizing Vectors.** Lower bounds for DP algorithms that can output a binary vector that is close (say, in the Hamming distance) to the input are well-known.

**Lemma 8** (e.g., [32]). *Let $\varepsilon, \delta > 0, m \in \mathbb{N}$, let the input domain be $\{0, 1\}^m$ and let two vectors $\mathbf{z}, \mathbf{z}' \in \{0, 1\}^m$ be neighbors if and only if $\|\mathbf{z} - \mathbf{z}'\|_0 \leq 1$. Then, for any $(\varepsilon, \delta)$-DP algorithm $\mathcal{M} : \{0, 1\}^m \to \{0, 1\}^m$, we have $\mathbb{E}_{\mathbf{z} \sim \{0,1\}^m}[\|\mathcal{M}(\mathbf{z}) - \mathbf{z}\|_0] \geq e^{-\varepsilon} \cdot m \cdot 0.5 \cdot (1 - \delta)$.*

It is also simple to extend the lower bound for the case where the vector is not binary, as stated below. We defer the full proof to Appendix B.

**Lemma 9.** *Let $D, m$ be any positive integer such that $D \geq 2$, let the input domain be $[D]^m$ and let two vectors $\mathbf{z}, \mathbf{z}' \in [D]^m$ be neighbors if and only if $\|\mathbf{z} - \mathbf{z}'\|_0 \leq 1$. Then, for any $(\ln(D/2), 0.25)$-DP algorithm $\mathcal{M} : [D]^m \to [D]^m$, we have that $\mathbb{E}_{\mathbf{z} \sim [D]^m}[\|\mathcal{M}(\mathbf{z}) - \mathbf{z}\|_0] \geq \Omega(m)$.*

**Group Differential Privacy.** For any neighboring relation $\sim$, we write $\sim_k$ as a neighboring relation where $D \sim_k D'$ iff there is a sequence $D = D_0, \ldots, D_{k'} = D'$ for some $k' \leq k$ such that $D_{i-1} \sim D_i$ for all $i \in [k']$.

**Fact 10** (e.g., [41]). *Let $\varepsilon > 0, \delta \geq 0$ and $k \in \mathbb{N}$. Suppose that $\mathcal{M}$ is an $(\varepsilon, \delta)$-DP algorithm for the neighboring relation $\sim$. Then $\mathcal{M}$ is $\left(k\varepsilon, \tfrac{e^{k\varepsilon} - 1}{e^{\varepsilon} - 1} \cdot \delta\right)$-DP for the neighboring relation $\sim_k$.*

## 3 DP Isotonic Regression over Total Orders

We first focus on the "one-dimensional" case where $\mathcal{X}$ is totally ordered; for convenience, we assume that $\mathcal{X} = [m]$ where the order is the natural order on integers. We first present an efficient algorithm for the this case and then a matching lower bound.

### 3.1 An Efficient Algorithm

To describe our algorithm, it will be more convenient to use the unnormalized version of the empirical risk, which we define as $\mathcal{L}^{\mathrm{abs}}(f; D) := \sum_{(x,y) \in D} \ell(f(x), y)$.

We now provide a high-level overview of our algorithm. Any monotone function $f : [m] \to [0, 1]$ contains a (not necessarily unique) threshold $\alpha \in \{0\} \cup [m]$ such that $f(a) \geq 1/2$ for all $a > \alpha$ and $f(a) \leq 1/2$ for all $a \leq \alpha$. Our algorithm works by first choosing this threshold $\alpha$ in a private manner using the exponential mechanism. The choice of $\alpha$ partitions $[m]$ into $[m]^{>\alpha} := \{a \in [m] \mid a > \alpha\}$ and $[m]^{\leq \alpha} := \{a \in [m] \mid a \leq \alpha\}$. The algorithm recurses on these two parts to find functions $f_> : [m]^{>\alpha} \to [1/2, 1]$ and $f_\leq : [m]^{\leq \alpha} \to [0, 1/2]$, which are then glued to obtain the final function.

In particular, the algorithm proceeds in $T$ stages, where in stage $t$, the algorithm starts with a partition of $[m]$ into $2^t$ intervals $\{P_{i,t} \mid i \in \{0, \dots, 2^t - 1\}\}$, and the algorithm eventually outputs a monotone function $f$ such that $f(x) \in [i/2^t, (i+1)/2^t]$ for all $x \in P_{i,t}$. This partition is further refined for stage $t+1$ by choosing a threshold $\alpha_{i,t}$ in $P_{i,t}$ and partitioning $P_{i,t}$ into $P_{i,t}^{>\alpha_{i,t}}$ and $P_{i,t}^{\leq \alpha_{i,t}}$. In the final stage, the function $f$ is chosen to be the constant $i/2^{T-1}$ over $P_{i,T-1}$. Note that the algorithm may stop at $T = \Theta_\varepsilon(\log n)$ because the Lipschitzness of $\ell$ ensures that assigning each partition to the constant $i/2^{T-1}$ cannot increase the error by more than $L/2^T \leq O_\varepsilon(L/n)$.

We already have mentioned above that each $\alpha_{i,t}$ has to be chosen in a private manner. However, if we let the scoring function directly be the unnormalized empirical risk, then its sensitivity remains as large as $L$ even at a large stage $t$. This is undesirable because the error from each run of the exponential mechanism can be as large as $O(L \cdot \log m)$ but there are as many as $2^t$ runs in stage $t$. Adding these error terms up would result in a far larger total error than desired.

To circumvent this, we observe that while the sensitivity can still be large, they are mostly "ineffective" because the function range is now restricted to only an interval of length $1/2^t$. Indeed, we may use the following "clipped" version of the loss function which has low sensitivity of $L/2^t$ instead.

**Definition 11** (Clipped Loss Function). *For a range $[\tau, \theta] \subseteq [0, 1]$, let $\ell_{[\tau,\theta]} : [\tau, \theta] \times [0, 1] \to \mathbb{R}$ be given as $\ell_{[\tau,\theta]}(\hat{y}, y) := \ell(\hat{y}, y) - \min_{y' \in [\tau,\theta]} \ell(y', y)$. Similar to above, we also define $\mathcal{L}^{\mathrm{abs}}_{[\tau,\theta]}(f; D) := \sum_{(x,y) \in D} \ell_{[\tau,\theta]}(f(x_i), y_i)$.*

Observe that $\mathrm{range}(\ell_{[\tau,\theta]}) \subseteq [0, L \cdot (\theta - \tau)]$, since $\ell$ is $L$-Lipschitz. In other words, the sensitivity of $\mathcal{L}^{\mathrm{abs}}_{[\tau,\theta]}(f; D)$ is only $L \cdot (\theta - \tau)$. Algorithm 1 contains a full description.

*Proof of Theorem 3.* Before proceeding to prove the algorithm's privacy and utility guarantees, we note that the output $f$ is indeed monotone since for every $x' < x$ that gets separated when we partition $P_{i,t}$ into $P_{2i,t+1}, P_{2i+1,t+1}$, we must have $x' \in P_{2i,t+1}$ and $x \in P_{2i+1,t+1}$.

**Privacy Analysis.** Since the exponential mechanism is $\varepsilon'$-DP and the dataset is partitioned with the exponential mechanism being applied only to each partition once, the parallel composition property (Lemma 6) implies that the entire subroutine for each $t$ is $\varepsilon'$-DP. Thus, by basic composition (Lemma 6), it follows that Algorithm 1 is $\varepsilon$-DP (since $\varepsilon = \varepsilon'T$).

**Utility Analysis.** Since the sensitivity of $\mathrm{score}_{i,t}(\cdot)$ is at most $L/2^t$, we have from Lemma 7, that for all $t \in \{0, \dots, T-1\}$ and $i \in \{0, 1, \dots, 2^t\}$,

$$\mathbb{E}\left[\mathrm{score}_{i,t}(\alpha_{i,t}) - \min_{\alpha \in P_{i,t}} \mathrm{score}_{i,t}(\alpha)\right] \leq O\left(\frac{L \cdot \log |P_{i,t}|}{\varepsilon' \cdot 2^t}\right) \leq O\left(\frac{L \cdot \log m}{\varepsilon' \cdot 2^t}\right). \quad (1)$$

Let $h_{i,t}$ denote $\mathrm{argmin}_{h \in \mathcal{F}(P_{i,t}, [i/2^t, (i+1)/2^t])} \mathcal{L}^{\mathrm{abs}}(h; D_{i,t})$ (with ties broken arbitrarily). Then, let $\tilde{\alpha}_{i,t}$ denote the largest element in $P_{i,t}$ such that $h_{i,t}(\tilde{\alpha}_{i,t}) \leq (i+1/2)/2^t$; namely, $\tilde{\alpha}_{i,t}$ is the optimal threshold when restricted to $D_{i,t}$. Under this notation, we have that

$$\mathrm{score}_{i,t}(\alpha_{i,t}) - \min_{\alpha \in P_{i,t}} \mathrm{score}_{i,t}(\alpha)$$

$$\geq \mathrm{score}_{i,t}(\alpha_{i,t}) - \mathrm{score}_{i,t}(\tilde{\alpha}_{i,t})$$

**Algorithm 1** DP Isotonic Regression for Totally Ordered Sets.

---

**Input:** $\mathcal{X} = [m]$, dataset $D = \{(x_1, y_1), \dots, (x_n, y_n)\}$, DP parameter $\varepsilon$.
**Output:** Monotone function $f : [m] \to [0, 1]$.

$T \leftarrow \lceil \log(\varepsilon n) \rceil$
$\varepsilon' \leftarrow \varepsilon/T$
$P_{0,0} \leftarrow [m]$
**for** $t = 0, \dots, T - 1$ **do**
  **for** $i = 0, \dots, 2^t - 1$ **do**
    ▷ $D_{i,t} \leftarrow \{(x_j, y_j) \mid j \in [n], x_j \in P_{i,t}\}$    {Set of all input points whose $x$ belongs to $P_{i,t}$}
      {Notation: Define $D_{i,t}^{\leq \alpha} := \{(x, y) \in D_{i,t} \mid x \leq \alpha\}$ and $D_{i,t}^{> \alpha}$ similarly }
      {Notation: Define $P_{i,t}^{\leq \alpha} := \{x \in P_{i,t} \mid x \leq \alpha\}$ and $P_{i,t}^{> \alpha}$ similarly }
    ▷ Choose threshold $\alpha_{i,t} \in \{0\} \cup P_{i,t}$, using $\varepsilon'$-DP exponential mechanism with scoring function

$$
\text{score}_{i,t}(\alpha) := \min_{f_1 \in \mathcal{F}(P_{i,t}^{\leq \alpha}, [\frac{i}{2^t}, \frac{i+0.5}{2^t}])} \mathcal{L}^{\text{abs}}_{[\frac{i}{2^t}, \frac{i+1}{2^t}]}(f_1; D_{i,t}^{\leq \alpha})
$$
$$
+ \min_{f_2 \in \mathcal{F}(P_{i,t}^{> \alpha}, [\frac{(i+0.5)}{2^t}, \frac{(i+1)}{2^t}])} \mathcal{L}^{\text{abs}}_{[\frac{i}{2^t}, \frac{i+1}{2^t}]}(f_2; D_{i,t}^{> \alpha})
$$

      {Note: $\text{score}_{i,t}(\alpha)$ has sensitivity at most $L/2^t$. }
    ▷ $P_{2i,t+1} \leftarrow P_{i,t}^{\leq \alpha_{i,t}}$ and $P_{2i+1,t+1} \leftarrow P_{i,t}^{> \alpha_{i,t}}$.
Let $f : [m] \to [0, 1]$ be given as $f(x) = i/2^{T-1}$ for all $x \in P_{i,T-1}$ and all $i \in [2^T]$.
**return** $f$

---

$$
= \left( \mathcal{L}^{\text{abs}}_{[i/2^t,(i+1)/2^t]}(h_{2i,t+1}; D_{2i,t+1}) + \mathcal{L}^{\text{abs}}_{[i/2^t,(i+1)/2^t]}(h_{2i+1,t+1}; D_{2i+1,t+1}) \right)
$$
$$
- \mathcal{L}^{\text{abs}}_{[i/2^t,(i+1)/2^t]}(h_{i,t}; D_{i,t})
$$
$$
= \mathcal{L}^{\text{abs}}(h_{2i,t+1}; D_{2i,t+1}) + \mathcal{L}^{\text{abs}}(h_{2i+1,t+1}; D_{2i+1,t+1}) - \mathcal{L}^{\text{abs}}(h_{i,t}; D_{i,t}). \tag{2}
$$

Finally, notice that

$$
\mathcal{L}^{\text{abs}}(f; D_{i,T-1}) - \mathcal{L}^{\text{abs}}(h_{i,T-1}; D_{i,T-1}) \leq \frac{L}{2^{T-1}} \cdot |D_{i,T-1}| = O\left( \frac{L \cdot |D_{i,T-1}|}{\varepsilon n} \right). \tag{3}
$$

With all the ingredients ready, we may now bound the expected (unnormalized) excess risk:

$$
\mathcal{L}^{\text{abs}}(f; D) = \sum_{0 \leq i < 2^{T-1}} \mathcal{L}^{\text{abs}}(f; D_{i,T-1})
$$
$$
\overset{(3)}{\leq} \sum_{0 \leq i < 2^{T-1}} \left( O\left( \frac{L \cdot |D_{i,T-1}|}{\varepsilon n} \right) + \mathcal{L}^{\text{abs}}(h_{i,T-1}; D_{i,T-1}) \right)
$$
$$
= O(L/\varepsilon) + \sum_{0 \leq i < 2^{T-1}} \mathcal{L}^{\text{abs}}(h_{i,T-1}; D_{i,T-1})
$$
$$
= O(L/\varepsilon) + \mathcal{L}^{\text{abs}}(h_{0,0}; D_{0,0})
$$
$$
+ \sum_{\substack{t \in [T-1] \\ 0 \leq i < 2^{t-1}}} \left( \mathcal{L}^{\text{abs}}(h_{2i,t}; D_{2i,t}) + \mathcal{L}^{\text{abs}}(h_{2i+1,t}; D_{2i+1,t}) - \mathcal{L}^{\text{abs}}(h_{i,t-1}; D_{i,t-1}) \right).
$$

Taking the expectation on both sides and using (1) and (2) yields

$$
\mathbb{E}[\mathcal{L}^{\text{abs}}(f; D)] \leq O(L/\varepsilon) + \mathcal{L}^{\text{abs}}(h_{0,0}; D_{0,0}) + \sum_{\substack{t \in [T-1] \\ 0 \leq i < 2^{t-1}}} O\left( \frac{L \cdot \log m}{\varepsilon' \cdot 2^t} \right)
$$
$$
= O(L/\varepsilon) + \mathcal{L}^{\text{abs}}(f^*; D) + O\left( T^2 \cdot \frac{L \cdot \log m}{\varepsilon} \right)
$$
$$
= \mathcal{L}^{\text{abs}}(f^*; D) + O\left( \frac{L \cdot \log m \cdot (1 + \log^2(\varepsilon n))}{\varepsilon} \right).
$$

Dividing both sides by $n$ yields the desired claim.

**Running Time.** To obtain a bound on the running time for general loss functions, we need to make a slight modification to the algorithm: we will additionally only restrict the range of $f_1, f_2$ to multiples of $1/2^{T-1}$. We remark that this does not affect the utility since anyway we always take the final output whose values are multiples of $1/2^{T-1}$.

Given any dataset $D = \{(x_1, y_1), \ldots, (x_n, y_n)\}$ where $x_1 < \cdots < x_n$, the *prefix isotonic regression* algorithm is to compute, for each $i \in [n]$, the optimal loss in isotonic regression on $(x_1, y_1), \ldots, (x_i, y_i)$. Straightforward dynamic programming solves this in $O(n \cdot v)$ time, where $v$ denote the number of possible values allowed in the function.

Now, for each $i, t$, we may run the above algorithm with $D = D_{i,t}$ and the allowed values are all multiples of $1/2^{T-1}$ in $[\frac{i}{2^t}, \frac{i+0.5}{2^t}]$; this gives us $\min_{f_1 \in \mathcal{F}(P_{i,t}^{\leq \alpha}, [\frac{i}{2^t}, \frac{i+0.5}{2^t}])} \mathcal{L}^{\text{abs}}_{[\frac{i}{2^t}, \frac{i+1}{2^t}]}(f_1; D_{i,t}^{\leq \alpha})$ for all $\alpha \in P_{i,t}$ in time $O(|D_{i,t}| \cdot 2^{T-t} + |P_{i,t}|)$. Analogously, we can also compute $\min_{f_2 \in \mathcal{F}(P_{i,t}^{> \alpha}, [\frac{(i+0.5)}{2^t}, \frac{(i+1)}{2^t}])} \mathcal{L}^{\text{abs}}_{[\frac{i}{2^t}, \frac{i+1}{2^t}]}(f_2; D_{i,t}^{> \alpha})$ for all $\alpha \in P_{i,t}$ in a similar time. Thus, we can compute $(\text{score}_{i,t}(\alpha))_{\alpha \in P_{i,t}}$ in time $O(|D_{i,t}| \cdot 2^{T-t} + |P_{i,t}|)$, and then sample accordingly.

We can further speed up the algorithm by observing that the score remains constant for all $\alpha \in [x_i, x_{i+1})$. Hence, we may first sample an interval among $[0, x_1), [x_1, x_2), \ldots, [x_{n-1}, x_n), [x_n, m)$ and then sample $\alpha_{i,t}$ uniformly from that interval. This entire process can be done in $O(|D_{i,t}| \cdot 2^{T-t} + \log m)$ time. In total, the running time of the algorithm is thus

$$\sum_{t=0}^{T-1} \sum_{i=0}^{2^t-1} O(|D_{i,t}| \cdot 2^{T-t} + \log m) \leq \sum_{t=0}^{T-1} O(n2^T + 2^t \cdot \log m) \leq O(n^2 \log n + n \log m).$$

**Near-Linear Time Algorithms for $\ell_1$-, $\ell_2^2$-Losses.** We now describe faster algorithms for the $\ell_1$- and $\ell_2^2$-loss functions, thereby proving the last part of Theorem 3. The key observation is that for convex loss functions, the restricted optimal is simple: we just have to "clip" the optimal function to be in the range $[\tau, \theta]$. Below $\text{clip}_{[\tau, \theta]}$ denotes the function $y \mapsto \min\{\theta, \max\{\tau, y\}\}$.

**Observation 12.** *Let $\ell$ be any convex loss function, $D$ any dataset, $f^* \in \text{argmin}_{f \in \mathcal{F}(\mathcal{X}, \mathcal{Y})} \mathcal{L}(f; D)$ and $\tau \leq \theta$ any real numbers such that $\tau, \theta \in \mathcal{Y}$. Define $f^*_{\text{clipped}}(x) := \text{clip}_{[\tau, \theta]}(f^*(x))$. Then, we must have $f^*_{\text{clipped}}(x) \in \text{argmin}_{f \in \mathcal{F}(\mathcal{X}, \mathcal{Y} \cap [\tau, \theta])} \mathcal{L}(f; D)$.*

*Proof.* Consider any $f \in \mathcal{F}(\mathcal{X}, \mathcal{Y} \cap [\tau, \theta])$. Let $\mathcal{X}^>$ (resp. $\mathcal{X}^<$) denote the set of all $x \in \mathcal{X}$ such that $f^*(x) > \theta$ (resp. $f^*(x) < \tau$). Consider the following operations:

- For each $x \in \mathcal{X}^>$, change $f(x)$ to $\theta$.
- For each $x \in \mathcal{X}^<$, change $f(x)$ to $\tau$.
- Let $f(x) = f^*(x)$ for all $x \in \mathcal{X} \setminus (\mathcal{X}^> \cup \mathcal{X}^<)$.

At the end, we end up with $f(x) = f^*_{\text{clipped}}(x)$. Each of the first two changes does not increase the loss $\mathcal{L}(f; D)$; otherwise, due to convexity, changing $f^*(x)$ to $f(x)$ would have decrease the objective function. Finally, the last operation does not decrease the loss; otherwise, we may replace this section of $f^*$ with the values in $f$ instead. Thus, we can conclude that $f^*_{\text{clipped}}(x) \in \text{argmin}_{g \in \mathcal{F}(\mathcal{X}, \mathcal{Y} \cap [\tau, \delta])} \mathcal{L}(f; D)$. $\square$

We will now show how to compute the scores in Algorithm 1 simultaneously for all $\alpha$ (for fixed $i, t$) in nearly linear time. To do this, recall the prefix isotonic regression problem from earlier. For this problem, Stout [42] gave an $O(n)$-time algorithm for $\ell_2$-loss and an $O(n \log n)$-time algorithm for $\ell_1$-loss (both the unrestricted value case). Furthermore, after the $i$th iteration, the algorithm also keeps a succinct representation $S_i^{\text{opt}}$ of the optimal solution in the form of an array $(i_1, v_1, \ell_1), \ldots, (i_k, v_k, \ell_k)$, which denotes $f(x) = v_j$ for all $x \in [x_{i_j}, x_{i_{j+1}})$, and $\ell_j$ indicates the loss $\mathcal{L}^{\text{abs}}$ up until $x_{i_{j+1}}$, not including.

We can extend the above algorithm to *prefix clipped isotonic regression* problem, which we define in the same manner as above except that we restrict the function range to be $[\tau, \theta]$ for some given $\tau < \theta$. Using Observation 12, it is not hard to extend the above algorithm to work in this case.

**Lemma 13.** *There is an $O(n \log n)$-time algorithm for $\ell_2^2$- and $\ell_1$-prefix clipped isotonic regression.*

*Proof.* We first precompute $c_\tau(i) = \sum_{j \leq i} \ell(\tau, x_j)$ and $c_\theta(i) = \sum_{j \leq i} \ell(\theta, x_j)$ for all $i \in [n]$. We then run the aforementioned algorithm from [42]. At each iteration $i$, use binary search to find the largest index $j_\tau$ such that $v_{j_\tau} < \tau$ and the largest index $j_\theta$ such that $v_{j_\theta} < \theta$. Observation 12 implies that the optimal solution of the clipped version is simply the same as the unrestricted version except that we need to change the function values before $x_{j_\tau}$ to $\tau$ and after $x_{j_\theta}$ to $\theta$. The loss of this clipped optimal can be written as $\ell_{j_\theta} - \ell_{j_\tau} + c_\tau(j_\tau) + (c_\theta(i) - c_\theta(j_\theta))$, which can be computed in $O(1)$ time given that we have already precomputed $c_\tau, c_\theta$. The running time of the entire algorithm is thus the same as that of [42] together with the binary search time; the latter totals to $O(n \log n)$. □

Our fast algorithm for computing $(\text{score}_{i,t}(\alpha))_{\alpha \in P_{i,t}}$ first runs the above algorithm with $\tau = \frac{i}{2^t}, \theta = \frac{i+0.5}{2^t}$ and $D = D_{i,t}$; this gives us $\min_{f_1 \in \mathcal{F}(P_{i,t}^{\leq \alpha}, [\frac{i}{2^t}, \frac{i+0.5}{2^t}])} \mathcal{L}_{[\frac{i}{2^t}, \frac{i+1}{2^t}]}^{\text{abs}}(f_1; D_{i,t}^{\leq \alpha})$ for all $\alpha \in P_{i,t}$ in time $O(|D_{i,t}| \log |D_{i,t}| + |P_{i,t}|)$. Analogously, we can also compute $\min_{f_2 \in \mathcal{F}(P_{i,t}^{>\alpha}, [\frac{(i+0.5)}{2^t}, \frac{(i+1)}{2^t}])} \mathcal{L}_{[\frac{i}{2^t}, \frac{i+1}{2^t}]}^{\text{abs}}(f_2; D_{i,t}^{>\alpha})$ for all $\alpha \in P_{i,t}$ in a similar time. Thus, we can compute $(\text{score}_{i,t}(\alpha))_{\alpha \in P_{i,t}}$ in time $O(|D_{i,t}| \log |D_{i,t}| + |P_{i,t}|)$, and sample accordingly. Using the same observation as the general loss function case, this can be sped up further to $O(|D_{i,t}| \log |D_{i,t}| + \log m)$ time. In total, the running time of the algorithm is thus

$$\sum_{t=0}^{T-1} \sum_{i=0}^{2^t-1} O(|D_{i,t}| \log |D_{i,t}| + \log m) \leq \sum_{t=0}^{T-1} O(n \log n + 2^t \log m) \leq O(n(\log^2 n + \log m)). \quad \square$$

## 3.2 A Nearly Matching Lower Bound

We show that the excess empirical risk guarantee in Theorem 3 is tight, even for approximate-DP algorithms with a sufficiently small $\delta$, under a mild assumption about the loss function stated below.

**Definition 14** (Distance-Based Loss Function). *For $R \geq 0$, a loss function $\ell$ is said to be $R$-distance-based if there exist $g : [0,1] \to \mathbb{R}_+$ such that $\ell(y, y') = g(|y - y'|)$ where $g$ is a non-decreasing function with $g(0) = 0$ and $g(1/2) \geq R$.*

We remark that standard loss functions, including $\ell_1$- or $\ell_2^2$-loss, are all $\Omega(1)$-distance-based.

Our lower bound is stated below. It is proved via a packing argument [25] in a similar manner as a lower bound for properly PAC learning threshold functions [10]. This is not a coincidence: indeed, when we restrict the range of our function to $\{0, 1\}$, the problem becomes exactly (the empirical version of) properly learning threshold functions. As a result, the same technique can be used to prove a lower bound in our setting as well.

**Theorem 15.** *For all $0 \leq \delta < 0.1 \cdot (e^\varepsilon - 1)/m$, any $(\varepsilon, \delta)$-DP algorithm for isotonic regression over $[m]$ for any $R$-distance-based loss function $\ell$ must have expected excess empirical risk $\Omega\left(R \cdot \min\left\{1, \frac{\log m}{\varepsilon n}\right\}\right)$.*

*Proof.* Suppose for the sake of contradiction that there exists an $(\varepsilon, \delta)$-DP algorithm $\mathcal{M}$ for isotonic regression with an $R$-distance-based loss function $\ell$ with expected excess empirical risk $0.01 \cdot \left(R \cdot \min\left\{1, \frac{\log(0.1m)}{\varepsilon n}\right\}\right)$. Let $k := \lfloor 0.1 \log(0.1m)/\varepsilon \rfloor$.

We may assume that $n \geq 2k$, as the $\Omega(R)$ lower bound for the case $n = 2k$ can easily be adapted for an $\Omega(R)$ lower bound for the case $n < 2k$ as well.

We will use the standard packing argument [25]. For each $j \in [m-1]$, we create a dataset $D_j$ that contains $k$ copies of $(j, 0)$, $k$ copies of $(j+1, 1)$ and $n - 2k$ copies of $(1, 0)$. Finally, let $D_m$ denote the dataset that contains $k$ copies of $(m, 0)$ and $n - k$ copies of $(1, 0)$. Let $V_j$ denote the set of all $f \in \mathcal{F}([m], [0, 1])$ such that $\mathcal{L}(f; \mathcal{D}) < Rk/n$. The utility guarantee of $\mathcal{M}$ implies that

$$\Pr[\mathcal{M}(D_j) \in V_j] \geq 0.5.$$

Furthermore, it is not hard to see that $V_1, \ldots, V_m$ are disjoint. In particular, for any function $f \in \mathcal{F}([m], [0, 1])$, let $x_f$ be the largest element $x \in [m]$ for which $f(x) \leq 1/2$; if no such $x$ exists (i.e., $f(0) > 1/2$), let $x_f = 0$. For any $j < x_f$, we have $\mathcal{L}(f; D_j) \geq \frac{k}{n} \ell(f(j+1), 1) \geq \frac{k}{n} \cdot g(1/2) \geq$

$Rk/n$. Similarly, for any $j > x_f$, we have $\mathcal{L}(f; D_j) \geq \frac{k}{n}\ell(f(j), 0)\frac{k}{n} \cdot g(1/2) \geq Rk/n$ This implies that $f$ can only belong to $V_j$, as claimed.

Therefore, we have that

$$
\begin{aligned}
1 &\geq \sum_{j \in [m]} \Pr[\mathcal{M}(D_m) \in V_j] \\
&\geq \sum_{j \in [m]} \frac{1}{e^{2k\varepsilon}} \left( \Pr[\mathcal{M}(D_j) \in V_j] - \delta \frac{(e^{2k\varepsilon} - 1)}{e^\varepsilon - 1} \right) \qquad \text{(Fact 10)} \\
&\geq \sum_{j \in [m]} \frac{10}{m}(0.5 - 0.1) \\
&> 1,
\end{aligned}
$$

a contradiction. □

## 3.3 Extensions

We now discuss several variants of the isotonic regression problem that places certain additional constraints on the function $f$ that we seek, as listed below.

- $k$-Piecewise Constant: $f$ must be a step function that consists of at most $k$ pieces.
- $k$-Piecewise Linear: $f$ must be a piecewise linear function with at most $k$ pieces.
- Lipschitz Regression: $f$ must be $L_f$-Lipschitz for some specified $L_f > 0$.
- Convex/Concave: $f$ must be convex/concave.

We devise a general meta algorithm that, with a small tweak in each case, works for all of these constraints to yield Theorem 4. At a high-level, our algorithm is similar to Algorithm 1, except that, in addition to using exponential mechanism to pick the threshold $\alpha_{i,t}$, we also pick certain auxiliary information that is then passed onto the next stage. For example, in the $k$-piecewise constant setting, the algorithm in fact picks also the number of pieces to the left of $\alpha_{i,t}$ and that to the right of it. These are then passed on to the next stage. The algorithm stops when the number of pieces become one, and then simply use the exponential mechanism to find the constant value on this subdomain.

The full description of the algorithm and the corresponding proof are deferred to Appendix C.

# 4 DP Isotonic Regression over General Posets

We now provide an algorithm and lower bounds for the case of general discrete posets. We first recall basic quantities about posets. An *anti-chain* of a poset $(\mathcal{X}, \leq)$ is a set of elements such that no two distinct elements are comparable, whereas a *chain* is a set of elements such that every pair of elements is comparable. The *width* of a poset $(\mathcal{X}, \leq)$, denoted by $\text{width}(\mathcal{X})$, is defined as the maximum size among all anti-chains in the poset. The *height* of $(\mathcal{X}, \leq)$, denoted by $\text{height}(\mathcal{X})$, is defined as the maximum size among all chains in the poset. Dilworth's theorem and Mirsky's theorem give the following relation between chains an anti-chains:

**Lemma 16** (Dilworth's and Mirsky's theorems [12, 36]). *A poset with width $w$ can be partitioned into $w$ chains. A poset with height $h$ can be partitioned into $h$ anti-chains.*

## 4.1 An Algorithm

Our algorithm for general posets is similar to that of totally ordered set presented in the previous section. The only difference is that, instead of attempting to pick a single maximal point $\alpha$ such that $f(\alpha) \leq \tau$ as in the previous case, there could now be many such maximal $\alpha$'s. Indeed, we need to use the exponential mechanism to pick all such $\alpha$'s. Since these are all maximal, they must be incomparable; therefore, they form an anti-chain. Since there can be as many as $|\mathcal{X}|^{\text{width}(\mathcal{X})}$ anti-chains in total, this means that the error from the exponential mechanism is $O\left(\log |\mathcal{X}|^{\text{width}(\mathcal{X})}/\varepsilon'\right) = O(\text{width}(\mathcal{X}) \log |\mathcal{X}|/\varepsilon')$, leading to the multiplicative increase of $\text{width}(\mathcal{X})$ in the total error. This completes our proof sketch for Theorem 1.

## 4.2 Lower Bounds

To prove a lower bound of $\Omega(\text{width}(\mathcal{X})/\varepsilon n)$, we observe that the values of the function in any anti-chain can be arbitrary. Therefore, we may use each element in a maximum anti-chain to encode $\mathcal{X}$

as a binary vector. The lower bound from Lemma 8 then gives us an $\Omega(\mathrm{width}(\mathcal{X})/n)$ lower bound for $\varepsilon = 1$, as formalized below.

**Lemma 17.** *For any $\delta > 0$, any $(1, \delta)$-DP algorithm for isotonic regression for any $R$-distance-based loss function $\ell$ must have expected excess empirical risk $\Omega\left(R(1-\delta) \cdot \min\left\{1, \frac{\mathrm{width}(\mathcal{X})}{n}\right\}\right)$.*

*Proof.* Consider any $(1, \delta)$-DP isotonic regression algorithm $\mathcal{M}'$ for loss $\ell$. Let $A$ be any maximum anti-chain (of size $\mathrm{width}(A)$) in $\mathcal{X}$. We use this algorithm to build a $(1, \delta)$-DP algorithm $\mathcal{M}$ for privatizing a binary vector of $m = \min\{n, |A| - 1\}$ dimensions as follows:

- Let $x_0, x_1, \ldots, x_m$ be distinct elements of $A$.
- On input $\mathbf{z} \in \{0, 1\}^m$, create a dataset $D = \{(x_1, z_1), \ldots, (x_m, z_m), (x_0, 0), \ldots, (x_0, 0)\}$ where $(x_0, 0)$ is repeated $n - m$ times.
- Run $\mathcal{M}'$ on the instance $D$ to get $f$, and output a vector $\mathbf{z}'$ where $z_i' = \mathbf{1}[f(x_i) \geq 1/2]$.

It is obvious that this algorithm is $(1, \delta)$-DP. Observe also that $\mathcal{L}(f^*; D) = 0$ and thus $\mathcal{M}'$'s expected excess empirical risk is $\mathbb{E}_{f \sim \mathcal{M}'(D)}[\mathcal{L}(f; D)] \geq R \cdot \mathbb{E}_{\mathbf{z}' \sim \mathcal{M}(\mathbf{z})}[\|\mathbf{z} - \mathbf{z}'\|_0]/n$, which, from Lemma 8, must be at least $\Omega(Rm(1-\delta)/n) = \Omega\left(R(1-\delta) \cdot \min\left\{1, \frac{\mathrm{width}(\mathcal{X})}{n}\right\}\right)$. $\square$

By using group privacy (Fact 10) and repeating each element $\Theta(1/\varepsilon)$ times, we arrive at a lower bound of $\Omega\left(R \cdot \min\left\{1, \frac{\mathrm{width}(\mathcal{X})}{\varepsilon n}\right\}\right)$. Furthermore, since $\mathcal{X}$ contains $\mathrm{height}(\mathcal{X})$ elements that form a totally ordered set, Theorem 15 gives a lower bound of $\Omega(R \cdot \log(\mathrm{height}(\mathcal{X}))/\varepsilon n)$ as long as $\delta < 0.01 \cdot \varepsilon / \mathrm{height}(\mathcal{X})$. Finally, due to Lemma 16, we have $\mathrm{height}(\mathcal{X}) \geq |\mathcal{X}| / \mathrm{width}(\mathcal{X})$, which means that $\max\{\mathrm{width}(\mathcal{X}), \log(\mathrm{height}(\mathcal{X}))\} \geq \Omega(\log|\mathcal{X}|)$. Thus, we arrive at:

**Theorem 18.** *For any $\varepsilon \in (0, 1]$ and any $\delta < 0.01 \cdot \varepsilon/|\mathcal{X}|$, any $(\varepsilon, \delta)$-DP algorithm for isotonic regression for $R$-distance-based loss function $\ell$ must have expected excess empirical risk $\Omega\left(R \cdot \min\left\{1, \frac{\mathrm{width}(\mathcal{X}) + \log|\mathcal{X}|}{\varepsilon n}\right\}\right)$.*

### 4.3 Tight Examples for Upper and Lower Bounds

Recall that our upper bound is $\tilde{O}\left(\frac{\mathrm{width}(\mathcal{X}) \cdot \log|\mathcal{X}|}{\varepsilon n}\right)$ while our lower bound is $\Omega\left(\frac{\mathrm{width}(\mathcal{X}) + \log|\mathcal{X}|}{\varepsilon n}\right)$. One might wonder whether this gap can be closed. Below we show that, unfortunately, this is impossible in general: there are posets for which each bound is tight.

**Tight Lower Bound Example.** Let $\mathcal{X}_{\mathrm{disj}(w,h)}$ denote the poset that consists of $w$ disjoint chains, $C_1, \ldots, C_w$ where $|C_1| = h$ and $|C_2| = \cdots = |C_w| = 1$. (Every pair of elements on different chains are incomparable.) In this case, we can solve the isotonic regression problem directly on each chain and piece the solutions together into the final output $f$. Note that $|\mathcal{X}_{\mathrm{disj}(w,h)}| = w + h - 1$ and $\mathrm{width}(\mathcal{X}) = w, \mathrm{height}(\mathcal{X}) = h$. According to Theorem 1, the unnormalized excess empirical risk in $C_i$ is $\tilde{O}(\log(|C_i|)/\varepsilon)$. Therefore, the total (normalized) empirical risk for the entire domain $\mathcal{X}$ is $\tilde{O}\left(\frac{\log h + (w-1)}{\varepsilon n}\right)$. This is at most $\tilde{O}\left(\frac{w}{\varepsilon n}\right)$ as long as $h \leq \exp(O(w))$; this matches the lower bound.

**Tight Upper Bound Example.** Consider the grid poset $\mathcal{X}_{\mathrm{grid}(w,h)} := [w] \times [h]$ where $(x, y) \leq (x', y')$ if and only if $x \leq x'$ and $y \leq y'$. We assume throughout that $w \leq h$. Observe that $\mathrm{width}(\mathcal{X}_{\mathrm{grid}(w,h)}) = w$ and $\mathrm{height}(\mathcal{X}_{\mathrm{grid}(w,h)}) = w + h$.

We will show the following lower bound, which matches the $\tilde{O}\left(\frac{\mathrm{width}(\mathcal{X}) \log|\mathcal{X}|}{\varepsilon n}\right)$ upper bound in the case where $h \geq w^{1 + \Omega(1)}$, up to $O(\log^2(\varepsilon n))$ factor. We prove it by a reduction from Lemma 9. Note that this reduction is in some sense a "combination" of the proofs of Theorem 15 and Lemma 17, as the coordinate-wise encoding aspect of Lemma 17 is still present (across the rows) whereas the packing-style lower bound is present in how we embed elements of $[D]$ (in blocks of columns).

**Lemma 19.** *For any $\varepsilon \in (0, 1]$ and $\delta < O_\varepsilon(1/h)$, any $(\varepsilon, \delta)$-DP algorithm for isotonic regression for any $R$-distance-based loss function $\ell$ must have expected excess empirical risk $\Omega\left(R \cdot \min\left\{1, \frac{w \cdot \log(h/w)}{\varepsilon n}\right\}\right)$.*

*Proof.* Let $D := \lfloor h/w - 1 \rfloor, m = w$ and $r := \min\{\lfloor 0.5n/m \rfloor, \lfloor 0.5\ln(D/2)/\varepsilon \rfloor\}$. Consider any $(\varepsilon, \delta)$-DP algorithm $\mathcal{M}'$ for isotonic regression for $\ell$ on $\mathcal{X}_{\mathrm{grid}(w,h)}$ where $\delta \leq 0.01\varepsilon/D$. We use this algorithm to build a $(\ln(D/2), 0.25)$-DP algorithm $\mathcal{M}$ for privatizing a vector $\mathbf{z} \in [D]^m$ as follows:

- Create a dataset $D$ that contains:
  - For all $i \in [m]$, $r$ copies of $((i, (w-i)(D+1) + z_i), 0)$ and $r$ copies of $((i, (w-i)(D+1) + z_i + 1), 1)$.
  - $n - 2rm$ copies of $((1,1), 0)$.
- Run $\mathcal{M}'$ on instance $D$ to get $f$.
- Output a vector $\mathbf{z}'$ where $z_i' = \max\{j \in [D] \mid f((i, (w-i)(D+1) + j)) \leq 1/2\}$. (For simplicity, when such $j$ does not exist let $z_i' = 0$.)

By group privacy, $\mathcal{M}$ is $(\ln(D/2), 0.25)$-DP. Furthermore, $\mathcal{L}(f^*; D) = 0$ and the expected empirical excess risk of $\mathcal{M}'$ is

$$
\mathbb{E}_{f \sim \mathcal{M}'(D)}[\mathcal{L}(f; D)]
$$
$$
\geq \tfrac{r}{n} \sum_{i \in [m]} \left( \ell(f(i, (w-i)(D+1) + z_i), 0) + \ell(f(i, (w-i)(D+1) + z_i + 1), 1) \right)
$$
$$
\geq \tfrac{r}{n} \sum_{i \in [m]} g(1/2) \cdot \mathbf{1}[z_i' \neq z_i] = \tfrac{Rr}{n} \cdot \|\mathbf{z} - \mathbf{z}'\|_0,
$$

which must be at least $\Omega(Rrm/n) = \Omega\left( R \cdot \min\left\{ 1, \frac{w \cdot \log(h/w)}{\varepsilon n} \right\} \right)$ by Lemma 9. $\square$

## 5 Additional Related Work

(Non-private) isotonic regression is well-studied in statistics and machine learning. The one-dimensional (aka univariate) case has long history [9, 46, 5, 44, 45, 13, 8, 35, 14, 15, 49]; for a general introduction, see [22]. Moreover, isotonic regression has been studied in higher dimensions [24, 27, 26], including the sparse setting [21], as well as in online learning [29]. A related line of work studies learning neural networks under (partial) monotonicity constraints [3, 50, 30, 40].

There has been a rich body of work on DP machine learning, including DP empirical risk minimization (ERM), e.g., [11, 6, 48, 47], and DP linear regression, e.g., [1]; however, to the best of our knowledge none of these can be applied to isotonic regression to obtain non-trivial guarantees.

Another line of work related to our setting is around privately learning threshold functions [7, 20, 10, 2, 28]. We leveraged this relation to prove our lower bound for totally ordered case (Section 3.2).

## 6 Conclusions

In this paper we obtained new private algorithms for isotonic regression on posets and proved nearly matching lower bounds in terms of the expected empirical excess risk. Although our algorithms for totally ordered sets are efficient, our algorithm for general posets is not. Specifically, a trivial implementation of the algorithm would run in time $\exp(\tilde{O}(\mathrm{width}(\mathcal{X})))$. It remains an interesting open question whether this can be sped up. To the best of our knowledge, this question does not seem to be well understood even for the non-private setting, as previous algorithmic works have focused primarily on the totally ordered case. Similarly, while our algorithm is efficient for the totally ordered sets, it remains interesting to understand whether nearly linear time algorithms for $\ell_1$- and $\ell_2^2$-losses can be extended to a larger class of loss functions.

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
