# A  Baseline Algorithm for Private Isotonic Regression

We provide a baseline algorithm for private isotonic regression by a direct application of the exponential mechanism. For simplicity, we start with the case of totally ordered sets and then extend the algorithm to general posets.

**Totally ordered sets.** Consider a discretized range of $\mathcal{T} := \left\{0, \frac{1}{T}, \frac{2}{T}, \ldots, 1\right\}$. We have that for $\tilde{f} := \operatorname{argmin}_{f \in \mathcal{F}([m], \mathcal{T})} \mathcal{L}(f; D)$ and $f^* := \operatorname{argmin}_{f \in \mathcal{F}([m], [0,1])} \mathcal{L}(f; D)$, it holds that $\mathcal{L}(\tilde{f}; D) \leq \mathcal{L}(f^*; D) + \frac{1}{T}$. Also, it is a simple combinatorial fact that $|\mathcal{F}([m], \mathcal{T})| = \binom{m+T}{T} \leq (m+T)^T$, which bounds the number of monotone functions with this discretization. Thus, the $\varepsilon$-DP exponential mechanism over the set of all monotone functions in $\mathcal{F}([m], \mathcal{T})$, with the score function $\mathcal{L}(f; D)$ of sensitivity at most $L/n$, returns $f : [m] \to \mathcal{T}$ such that

$$\mathcal{L}(f; D) \leq \mathcal{L}(\tilde{f}; D) + O\left(\frac{LT \log(m+T)}{\varepsilon n}\right) \leq \mathcal{L}(f^*; D) + O\left(\frac{LT \log(m+T)}{\varepsilon n} + \frac{L}{T}\right).$$

Setting $T = \sqrt{\frac{\varepsilon n}{\log m}}$, gives an excess empirical error of $O\left(L\sqrt{\frac{\log(m)}{\varepsilon n}}\right)$ (when $m \geq n$).

**General posets.** By Lemma 16, we have that $\mathcal{X}$ can be partitioned into $w := \operatorname{width}(\mathcal{X})$ many chains $H_1, \ldots, H_w$. Let $h_i := |H_i|$. Since any monotone function over $\mathcal{X}$ has to be monotone over each of the chains, we have that

$$|\mathcal{F}(\mathcal{X}, \mathcal{T})| \leq \prod_{i=1}^{w} |\mathcal{F}(H_i, \mathcal{T})| \leq \left(\frac{|\mathcal{X}|}{w} + T\right)^{wT} \leq (|\mathcal{X}| + T)^{wT}.$$

Thus, by a similar argument as above, the $\varepsilon$-DP exponential mechanism over the set of all monotone functions in $\mathcal{F}(\mathcal{X}, \mathcal{T})$, with score function $\mathcal{L}(f; D)$ returns $f : \mathcal{X} \to \mathcal{T}$ such that

$$\mathcal{L}(f; D) \leq \mathcal{L}(f^*; D) + O\left(L \cdot \frac{wT \log(|\mathcal{X}| + T)}{n\varepsilon} + \frac{L}{T}\right).$$

Choosing $T = \sqrt{\frac{\varepsilon n}{w \log |\mathcal{X}|}}$, gives an excess empirical error of $O\left(L\sqrt{\frac{w \log |\mathcal{X}|}{\varepsilon n}}\right)$ (when $|\mathcal{X}| \geq n$).

# B  Lower Bound on Privatizing Vectors with Large Alphabet: Proof of Lemma 9

Below we prove Lemma 9. The proof below is a slight extension of that of Lemma 8 in [32].

*Proof of Lemma 9.* For every $i \in [m], \sigma \in [D]$, let $\mathbf{z}_{(i,\sigma)}$ denote $(z_1, \ldots, z_{i-1}, \sigma, z_{i+1}, \ldots, z_m)$. Let $\varepsilon' = \ln(D/2), \delta' = 0.25$. We have

$$\mathbb{E}_{\mathbf{z} \sim [D]^m}[\|\mathcal{M}(\mathbf{z}) - \mathbf{z}\|_0]$$

$$= \sum_{i \in [m]} \Pr_{\mathbf{z} \sim [D]^m}[\mathcal{M}(\mathbf{z})_i \neq z_i]$$

$$= m - \sum_{i \in [m]} \Pr_{\mathbf{z} \sim [D]^m}[\mathcal{M}(\mathbf{z})_i = z_i]$$

$$= m - \sum_{i \in [m]} \frac{1}{D^{m+1}} \sum_{\mathbf{z} \in [D]^m} \left(\sum_{\sigma \in [D]} \Pr[\mathcal{M}(\mathbf{z}_{(i,\sigma)}) = \sigma]\right)$$

$$\text{(From } (\varepsilon', \delta')\text{-DP of } \mathcal{M}) \geq m - \sum_{i \in [m]} \frac{1}{D^{m+1}} \sum_{\mathbf{z} \in [D]^m} \left(\sum_{\sigma \in [D]} \left(e^{\varepsilon'} \cdot \Pr[\mathcal{M}(\mathbf{z}_{(i,1)}) = \sigma] + \delta'\right)\right)$$

$$\geq m - \sum_{i \in [m]} \frac{1}{D^{m+1}} \sum_{\mathbf{z} \in [D]^m} \left(e^{\varepsilon'} + D\delta'\right)$$

$$= \left(1 - e^{\varepsilon'}/D - \delta'\right) m$$

$$= 0.25m. \qquad \square$$

# C  Algorithms for Isotonic Regression with Additional Constraints

In this section, we elaborate on the constrained variants of the isotonic regression problem over totally ordered sets, by designing a meta-algorithm that can be instantiated to get algorithms for each of the cases discussed in Section 3.3.

Recall that Algorithm 1 proceeded in $T$ rounds where in round $t$ the algorithm starts with a partition of $[m]$ into $2^t$ intervals, and then partitions each interval into two using the exponential mechanism. At a high-level, our meta-algorithm is similar, except that, it maintains a set of pairwise disjoint *structured intervals* of $[m]$, that is, each interval has an additional structure which imposes constraints on the function that can be returned on the said interval; moreover, the function is fixed outside the union of the said intervals. This idea is described in Algorithm 2, stated using the following abstractions, which will be instantiated to derive algorithms for each constrained variant.

- A set of all *structured intervals* of $[m]$ denoted as $\mathcal{S}$, and an *initial structured interval* $S_{0,0} \in \mathcal{S}$. A structured interval $S$ will consist of an interval domain denoted $P_S \subseteq [m]$, an interval range denoted $R_S \subseteq [0,1]$, and potentially additional other constraints that the function should satisfy. We use $|R_S|$ to denote the length of $R_S$. In order to make the number of structured intervals bounded, we will consider a discretized range where the endpoints of interval $R_S$ lie in $\mathcal{H} := \{0, 1/H, 2/H, \ldots, 1\}$ for some discretization parameter $H$.
- A *partition method* $\Phi : S \mapsto \{(S^{\text{left}}, S^{\text{right}}, g)\}$ that defines a set of all "valid partitions" of a structured interval $S$ into two structured intervals $S^{\text{left}}$ and $S^{\text{right}}$ and a function $g : P_S \setminus (P_{S^{\text{left}}} \cup P_{S^{\text{right}}}) \to R_S$. It is required that $P_S \setminus (P_{S^{\text{left}}} \cup P_{S^{\text{right}}})$ be an interval. If the algorithm makes a choice of $(S^{\text{left}}, S^{\text{right}}, g)$, then the final function returned by the algorithm is required to be equal to $g$ on $P_S \setminus (P_{S^{\text{left}}} \cup P_{S^{\text{right}}})$.
- For all $S \in \mathcal{S}$, we abuse notation to let $\mathcal{F}(S)$ denote the set of all monotone functions mapping $P_S$ to $R_S$, while respecting the additional conditions enforced by the structure in $S$.

We instantiate this notion of structured intervals in the following ways to derive algorithms for the constrained variants of isotonic regression mentioned earlier:

- *(Vanilla) Isotonic Regression (recovers Algorithm 1)*: $\mathcal{S}$ is simply the set of all interval domains, and all (discretized) interval ranges and the partition method simply partitions into two sub-intervals, with the range divided into two equal parts.[2] Namely,

$$
\begin{aligned}
\mathcal{S} &:= \{([i,j],[\tau,\theta]) : i,j \in [m], \tau, \theta \in \mathcal{H} \text{ s.t. } i \le j, \tau \le \theta\}, \\
S_{0,0} &:= ([1,m],[0,1]), \\
\Phi(([i,j],[\tau,\theta])) &:= \{(([i,\ell],[\tau,\tfrac{\tau+\theta}{2}]),([\ell+1,j],[\tfrac{\tau+\theta}{2},\theta])) : i-1 \le \ell \le j\}, \\
\mathcal{F}(([i,j],[\tau,\theta])) &:= \text{set of monotone functions mapping } [i,j] \text{ to } [\tau,\theta].
\end{aligned}
$$

We skip the description of the function $g$ in the partition method $\Phi$, since the middle sub-interval is empty. For all the other variants, we skip having to explicitly write the conditions of $i,j \in [m]$, $\tau, \theta \in \mathcal{H}$, $i \le j$, and $\tau \le \theta$ in definition of $\mathcal{S}$, and similarly that $\mathcal{F}(S)$ consist of monotone functions mapping $[i,j]$ to $[\tau,\theta]$; we only focus on the main new conditions.

- *k-Piecewise Constant*: $\mathcal{S}$ is the set of all interval domains, all discretized ranges, along with a parameter (encoding an upper bound on the number of pieces in the final piecewise constant function). The partition method partitions into two sub-intervals respecting that the number of pieces and the range divided into two equal parts, namely,

$$
\begin{aligned}
\mathcal{S} &:= \left\{([i,j],[\tau,\theta],r) : \begin{array}{ll} 1 \le r \le k & \text{if } i \le j, \\ r = 0 & \text{if } i > j \end{array}\right\}, \\
S_{0,0} &:= ([1,m],[0,1],k), \\
\Phi(([i,j],[\tau,\theta],r)) &:= \left\{\begin{array}{l} (([i,\ell],[\tau,\tfrac{\tau+\theta}{2}],r_1),([\ell+1,j],[\tfrac{\tau+\theta}{2},\theta],r_2)) \\ \quad \text{s.t. } i-1 \le \ell \le j \text{ and } r_1 + r_2 = r \end{array}\right\}, \\
\mathcal{F}(([i,j],[\tau,\theta],r)) &:= \text{set of } r\text{-piecewise constant functions}
\end{aligned}
$$

---

[2]We ignore a slight detail that $\frac{\tau+\theta}{2}$ need not be in $\mathcal{H}$; this can be fixed e.g., by letting it be $\lfloor H \cdot \frac{\tau+\theta}{2} \rfloor / H$, but we skip this complicated expression for simplicity. Note that, if we let $H = 2^T$, this distinction does not make a difference in the algorithm for vanilla isotonic regression.

**Algorithm 2** Meta algorithm for variants of DP Isotonic Regression for Totally Ordered Sets.

---

**Input:** $\mathcal{X} = [m]$, dataset $D = \{(x_1, y_1), \ldots, (x_n, y_n)\}$, DP parameter $\varepsilon$.
**Output:** Monotone function $f : [m] \to [0, 1]$ satisfying additional desired condition.

$T \leftarrow \lceil \log(\varepsilon n) \rceil$
$\varepsilon' \leftarrow \varepsilon / T$
$S_{0,0}$ : initial structured interval $\qquad\qquad$ {Any structured interval $S$ consists of an interval domain $P_S$ and an interval range $R_S$, and potentially other conditions on the function.}
**for** $t = 0, \ldots, T - 1$ **do**
$\quad$ **for** $i = 0, \ldots, 2^t - 1$ **do**
$\quad\quad$ ▷ $D_{i,t} \leftarrow \{(x_j, y_j) \mid j \in [n], x_j \in P_{S_{i,t}}\}$
$\quad\quad$ ▷ Choose $(S^{\mathrm{left}}_{i,t}, S^{\mathrm{right}}_{i,t}, g_{i,t}) \in \Phi(S_{i,t})$, using $\varepsilon'$-DP exponential mechanism with scoring function

$$\mathrm{score}_{i,t}(S^{\mathrm{left}}, S^{\mathrm{right}}, g) := \min_{f_1 \in \mathcal{F}(S^{\mathrm{left}})} \mathcal{L}^{\mathrm{abs}}_{R_S}(f_1; D^{\mathrm{left}}_{i,t})$$
$$+ \min_{f_2 \in \mathcal{F}(S^{\mathrm{right}})} \mathcal{L}^{\mathrm{abs}}_{R_S}(f_2; D^{\mathrm{right}}_{i,t})$$
$$+ \mathcal{L}^{\mathrm{abs}}_{R_S}(g; D^{\mathrm{mid}}_{i,t})$$

$\quad\quad$ {Notation: $D^{\mathrm{left}}_{i,t} := \{(x, y) \in D_{i,t} \mid x \in P_{S^{\mathrm{left}}}\}$, $D^{\mathrm{right}}_{i,t}$ is defined similarly and
$\quad\quad\quad\quad D^{\mathrm{mid}}_{i,t} := \{(x, y) \in D_{i,t} \mid x \in P_{S_{i,t}} \smallsetminus (P_{S^{\mathrm{left}}} \cup P_{S^{\mathrm{right}}})\}$.}
$\quad\quad$ {Note: $\mathrm{score}_{i,t}(S^{\mathrm{left}}, S^{\mathrm{right}}, g)$ has sensitivity at most $L \cdot |R_S|$.}

$\quad\quad$ ▷ $S_{2i,t+1} \leftarrow S^{\mathrm{left}}_{i,t}$ and $S_{2i+1,t+1} \leftarrow S^{\mathrm{right}}_{i,t}$.

Let $f : [m] \to [0, 1]$ be choosing $f|_{P_{S_{i,T-1}}} \in \mathcal{F}(S_{i,T-1})$ arbitrarily for all $i \in [2^T]$, and $f(x) = g_{i,t}(x)$ for all $x \in P_{S_{i,t}} \smallsetminus (P_{S_{2i,t}} \cup P_{S_{2i+1,t}})$ for all $i, t$.
**return** $f$

---

- *k-Piecewise Linear*: $\mathcal{S}$ is the set of all interval domains, all discretized ranges, along with a parameter (encoding an upper bound on the number of pieces in the final piecewise linear function), and two Boolean values ($\top/\bot$), one encoding whether the function must achieve the minimum possible value at the start of the interval, and other encoding whether it must achieve the maximum possible value at the end of the interval. The partition method partitions into two sub-intervals respecting that the number of pieces, by choosing a middle sub-interval that ensures that each range is at most half as large as the earlier one, namely,

$$\mathcal{S} := \left\{ ([i, j], [\tau, \theta], r, b_1, b_2) : \begin{array}{ll} 1 \le r \le k & \text{if } i \le j, \\ r = 0 & \text{if } i > j, \\ b_1, b_2 \in \{\top, \bot\} & \end{array} \right\},$$

$$S_{0,0} := ([1, m], [0, 1], k, \bot, \bot),$$

$$\Phi(([i, j], [\tau, \theta], r, b_1, b_2)) := \left\{ \begin{array}{c} \left( \begin{array}{c} S^{\mathrm{left}} = ([i, \ell_1], [\tau, \omega_1], r_1, b_1, \top), \\ S^{\mathrm{right}} = ([\ell_2, j], [\omega_2, \theta], r_2, \top, b_2) \\ g(x) = \omega_1 + (x - \ell_1) \cdot (\omega_2 - \omega_1)/(\ell_2 - \ell_1) \end{array} \right) \\ \text{s.t. } i - 1 \le \ell_1 < \ell_2 \le j + 1, \omega_1 \le \frac{\tau + \theta}{2} \le \omega_2, \\ \text{and } r_1 + r_2 = r - 1 \end{array} \right\},$$

$$\mathcal{F}(([i, j], [\tau, \theta], r, b_1, b_2)) := \text{set of } r\text{-piecewise linear functions } f$$
$$\text{s.t. } f(i) = \tau \text{ if } b_1 = \top \text{ and } f(j) = \theta \text{ if } b_2 = \top.$$

In other words, $\Phi(([i, j], [\tau, \theta], r, b_1, b_2))$ considers the three sub-intervals $[i, \ell_1]$, $[\ell_1, \ell_2]$ and $[\ell_2, j]$, and fits an affine function $g$ in the middle sub-interval $[\ell_1, \ell_2]$ such that $g(\ell_1) = \omega_1$ and $g(\ell_2) = \omega_2$ and ensures that the function $f$ returned on sub-intervals $[i, \ell_1]$ and $[\ell_2, j]$ satisfies $f(\ell_1) = \omega_1$ and $f(\ell_2) = \omega_2$.
- *Lipschitz Regression*: Given any Lipschitz constant $L_f$, $\mathcal{S}$ is the set of all interval domains, all discretized ranges, along with two Boolean values ($\top/\bot$), one encoding whether the function

must achieve the minimum possible value at the start of the interval, and other encoding whether it must achieve the maximum possible value at the end of the interval. The partition method chooses sub-intervals by choosing $\ell$ and function values $f(\ell)$ and $f(\ell+1)$ such that $f(\ell+1) - f(\ell) \leq L_f$ (thereby respecting the Lipschitz condition), and moreover $f(\ell) \leq \frac{\tau+\theta}{2}$ and $f(\ell+1) \geq \frac{\tau+\theta}{2}$.

$$\mathcal{S} := \{([i,j],[\tau,\theta],b_1,b_2) : b_1, b_2 \in \{\top, \bot\}\},$$
$$S_{0,0} := ([1,m],[0,1],\bot,\bot),$$
$$\Phi(([i,j],[\tau,\theta],b_1,b_2)) := \left\{ \begin{pmatrix} S^{\text{left}} = ([i,\ell],[\tau,\omega_1],b_1,\top), \\ S^{\text{right}} = ([\ell+1,j],[\omega_2,\theta],\top,b_2) \end{pmatrix} \atop \text{s.t. } i-1 \leq \ell \leq j \, , \omega_1 \leq \frac{\tau+\theta}{2} \leq \omega_2 \, , \atop \omega_2 - \omega_1 \leq L_f \right\},$$
$$\mathcal{F}(([i,j],[\tau,\theta],b_1,b_2)) := \text{set of } L_f\text{-Lipschitz linear functions } f$$
$$\text{s.t. } f(i) = \tau \text{ if } b_1 = \top \text{ and } f(j) = \theta \text{ if } b_2 = \top.$$

- *Convex/Concave*: We only describe the convex case; the concave case follows similarly. Note that a function $f$ is convex over the discrete domain $[m]$ if and only if $f(x+1) + f(x-1) > 2 \cdot f(x)$ holds for all $x$. Let $\mathcal{S}$ be the set of all interval domains, all discretized ranges, along with the following additional parameters
  - a lower bound $L$ on the (discrete) derivative of $f$,
  - an upper bound $U$ on the (discrete) derivative of $f$,
  - a Boolean value encoding whether the function must achieve the minimum possible value at the start of the interval,
  - another Boolean value encoding whether the function must achieve the maximum possible value at the end of the interval.

The partition method chooses sub-intervals by choosing $\ell$ and function values $f(\ell)$ and $f(\ell+1)$ such that $L \leq f(\ell+1) - f(\ell) \leq U$, $f(\ell) \leq \frac{\tau+\theta}{2}$ and $f(\ell+1) \geq \frac{\tau+\theta}{2}$ and enforcing that the left sub-interval has derivatives at most $f(\ell+1) - f(\ell)$ and the right sub-interval has derivatives at least $f(\ell+1) - f(\ell)$.

$$\mathcal{S} := \{([i,j],[\tau,\theta],L,U,b_1,b_2) : L \leq U, b_1, b_2 \in \{\top, \bot\}\},$$
$$S_{0,0} := ([1,m],[0,1],-\infty,+\infty,\bot,\bot),$$
$$\Phi(([i,j],[\tau,\theta],L,U,b_1,b_2)) := \left\{ \begin{pmatrix} S^{\text{left}} = ([i,\ell],[\tau,\omega_1],L,\omega_2-\omega_1,b_1,\top), \\ S^{\text{right}} = ([\ell+1,j],[\omega_2,\theta],\omega_2-\omega_1,U,\top,b_2) \end{pmatrix} \atop \text{s.t. } i-1 \leq \ell \leq j \, , \omega_1 \leq \frac{\tau+\theta}{2} \leq \omega_2 \, , \atop L \leq \omega_2 - \omega_1 \leq U \right\},$$
$$\mathcal{F}(([i,j],[\tau,\theta],L,U)) := \text{set of convex functions } f$$
$$\text{s.t. for all } \ell \in [i,j) \text{ it holds that } L \leq f(\ell+1) - f(\ell) \leq U,$$
$$\text{and } f(i) = \tau \text{ if } b_1 = \top \text{ and } f(j) = \theta \text{ if } b_2 = \top.$$

**Privacy Analysis.** Follows similarly as done for Algorithm 1.

**Utility Analysis.** Since $|R_{S_{i,t}}| \leq 2^{-t}$ in each of the cases, it follows that the sensitivity of the scoring function is at most $L/2^t$. The rest of the proof follows similarly, with the only change being that the number of candidates in the exponential mechanism is given as $|\Phi(S_{i,t})|$, which in the case of vanilla isotonic regression was simply $|P_{i,t}|$. We now bound this for each of the cases, which shows that $\log |\Phi(S_{i,t})|$ is at most $O(\log(mn))$. In particular,

- *k-Piecewise Constant:* $|\Phi(S)| \leq O(mk)$.
- *k-Piecewise Linear:* $|\Phi(S)| \leq O(m^2 H^2 k)$.
- *$L_f$-Lipschitz:* $|\Phi(S)| \leq O(mH^2)$.
- *Convex/Concave:* $|\Phi(S)| \leq O(mH)$

Finally, there is an additional error due to discretization. To account for the discretization error, we argue below for appropriately selected values of $H$ that, for any optimal function $f^*$, there exists $f \in \mathcal{F}(S_{0,0})$ such that $|f^*(x) - f(x)| \leq 1/n$. This indeed immediately implies that the discretization error is at most $O(1)$.

- **$k$-Piecewise Linear:** We may select $H = n$. In this case, for every endpoint $\ell$, we let $f(\ell) = H \cdot \lceil f^*(\ell)/H \rceil$ and interpolate the intermediate points accordingly. It is simple to see that $f^*(x) - f(x) \leq 1/n$ as desired.
- **$L_f$-Lipschitz and *Convex/Concave*:** Let $H = mn$. Here we discretize the (discrete) derivative of $f$. Specifically, let $f(1) = \lfloor H \cdot f^*(1) \rfloor / H$ and let $f(\ell+1) - f(\ell) = \lfloor H \cdot (f^*(\ell+1) - f^*(\ell)) \rfloor / H$ for all $\ell = 2, \ldots, m$. Once again, it is simple to see that $f, f^*$ differ by at most $1/n$ at each point.

In summary, in all cases, we have $|\Phi(S)| \leq (nm)^{O(1)}$ resulting in the same asymptotic error as in the unconstrained case.

**Runtime Analysis.** It is easy to see that each score value can be computed (via dynamic programming) in time $\mathrm{poly}(n) \cdot \mathrm{poly}(H)$. Thus, the entire algorithm can be implemented in time that $\mathrm{poly}(n) \cdot \mathrm{poly}(H) \cdot \log m \leq (nm)^{O(1)}$ as claimed.[3]

# D  Missing Proofs from Section 4

For a set $S \subseteq \mathcal{X}$, its lower closure and upper closure are defined as $S^{\leq} := \{x \in \mathcal{X} \mid \exists s \in S, x \leq s\}$ and $S^{\geq} := \{x \in \mathcal{X} \mid \exists s \in S, x \geq s\}$, respectively. Similarly, the strict lower closure and strict upper closure are defined as $S^{<} := \{x \in \mathcal{X} \mid \exists s \in S, x < s\}$ and $S^{>} := \{x \in \mathcal{X} \mid \exists s \in S, x > s\}$. When $S = \emptyset$, we use the convention that $S^{\leq} = S^{<} = \emptyset$ and $S^{\geq} = S^{>} = \mathcal{X}$.

## D.1  Proof of Theorem 1

We note that, in the proof below, we also consider the empty set to be an anti-chain.

*Proof of Theorem 1.* We use the notations of $\ell_{[\tau,\theta]}$ and $\mathcal{L}^{\mathrm{abs}}_{[\tau,\theta]}$ as defined in the proof of Theorem 3.

Any monotone function $f : \mathcal{X} \to [0,1]$ corresponds to an antichain $A$ in $\mathcal{X}$ such that $f(a) \geq 1/2$ for all $a \in A^{>}$ and $f(a) \leq 1/2$ for all $a \in A^{\leq}$. Our algorithm works by first choosing this antichain $A$ in a DP manner using the exponential mechanism. The choice of $A$ partitions the poset into two parts $A^{>}$ and $A^{\leq}$ and the algorithm recurses on these two parts to find functions $f_{>} : A^{>} \to [1/2, 1]$ and $f_{\leq} : A^{\leq} \to [0, 1/2]$, which are put together to obtain the final function.

In particular, the algorithm proceeds in $T$ stages, where in stage $t$, the algorithm starts with a partition of $\mathcal{X}$ into $2^t$ parts $\{P_{i,t} \mid i \in [2^t]\}$, and the algorithm eventually outputs a monotone function $f$ such that $f(x) \in [i/2^t, (i+1)/2^t]$ for all $x \in P_{i,t}$. This partition is further refined for stage $t+1$ by choosing an antichain $A_{i,t}$ in $P_{i,t}$ and partitioning $P_{i,t}$ into $P_{i,t} \cap A^{>}_{i,t}$ and $P_{i,t} \cap A^{\leq}_{i,t}$. In the final stage, the function $f$ is chosen to be the constant $i/2^{T-1}$ over $P_{i,T-1}$. A complete description is presented in Algorithm 3.

Before proceeding to prove the algorithm's privacy and utility guarantees, we note that the output $f$ is indeed monotone because for every $x' < x$ that gets separated when we partition $P_{i,t}$ to $P_{2i,t+1}, P_{2i+1,t+1}$, we must have $x' \in P_{2i,t+1}$ and $x \in P_{2i+1,t+1}$.

**Privacy Analysis.** Similar to the proof of Theorem 3, it follows that each inner subroutine for each $t$ is $\varepsilon'$-DP, and thus the entire mechanism is $\varepsilon$-DP by basic composition of DP (Lemma 6).

**Utility Analysis.** Since the sensitivity of $\mathrm{score}_{i,t}(\cdot)$ is at most $L/2^t$, we have from Lemma 7, that for all $t \in \{0, \ldots, T-1\}$ and $i \in [2^t]$,

$$\mathbb{E}\left[\mathrm{score}_{i,t}(A_{i,t}) - \min_{A \in \mathcal{A}_{i,t}} \mathrm{score}_{i,t}(A)\right] \leq O\left(\frac{L \cdot \log|\mathcal{A}_{i,t}|}{\varepsilon' \cdot 2^t}\right) \leq O\left(\frac{L \cdot \mathrm{width}(\mathcal{X}) \cdot \log|\mathcal{X}|}{\varepsilon' \cdot 2^t}\right). \tag{4}$$

To facilitate the subsequent steps of the proof, let us introduce additional notation. Let $h_{i,t}$ denote $\mathrm{argmin}_{h \in \mathcal{F}(P_{i,t}, [i/2^t, (i+1)/2^t])} \mathcal{L}^{\mathrm{abs}}(h; D_{i,t})$ (with ties broken arbitrarily). Then, let $\tilde{A}_{i,t}$ denote the

---

[3]In the main body, we erroneously claimed that the running time was $(n \log m)^{O(1)}$, instead of $(nm)^{O(1)}$.

**Algorithm 3** DP Isotonic Regression for General Posets

---

**Input:** Poset $\mathcal{X}$, dataset $D = \{(x_1, y_1), \ldots, (x_n, y_n)\}$, DP parameter $\varepsilon$.
**Output:** Monotone function $f : \mathcal{X} \to [0, 1]$.

$T \leftarrow \lceil \log(\varepsilon n) \rceil$ and $\varepsilon' \leftarrow \varepsilon/T$.
$P_{0,0} \leftarrow \mathcal{X}$
**for** $t = 0, \ldots, T - 1$ **do**
    **for** $i = 0, \ldots, 2^t - 1$ **do**

        ▷ $D_{i,t} \leftarrow \{(x_j, y_j) \mid j \in [n], x_j \in P_{i,t}\}$ (set of all input points whose $x$ belongs to $P_{i,t}$)

        ▷ $\mathcal{A}_{i,t} \leftarrow$ set of all antichains in $P_{i,t}$.

        For each antichain $A \in \mathcal{A}_{i,t}$, we abuse notation to use
          • $D_{i,t} \cap A^{\leq}$ to denote $\{(x, y) \in D_{i,t} \mid x \in A^{\leq}\}$, and
          • $D_{i,t} \cap A^{>}$ to denote $\{(x, y) \in D_{i,t} \mid x \in A^{>}\}$.

        ▷ Choose antichain $A_{i,t} \in \mathcal{A}_{i,t}$ using the exponential mechanism with the scoring function

$$\text{score}_{i,t}(A) = \min_{f_1 \in \mathcal{F}(P_{i,t} \cap A^{\leq}, [\frac{i}{2^t}, \frac{i+0.5}{2^t}])} \mathcal{L}^{\text{abs}}_{[\frac{i}{2^t}, \frac{i+1}{2^t}]}(f_1; D_{i,t} \cap A^{\leq})$$
$$+ \min_{f_2 \in \mathcal{F}(P_{i,t} \cap A^{>}, [\frac{i+0.5}{2^t}, \frac{i+1}{2^t}])} \mathcal{L}^{\text{abs}}_{[\frac{i}{2^t}, \frac{i+1}{2^t}]}(f_2; D_{i,t} \cap A^{>}),$$

        $\{\text{score}_{i,t}(A) \text{ has sensitivity at most } L/2^t.\}$

        ▷ $P_{2i,t+1} \leftarrow P_{i,t} \cap A^{\leq}_{i,t}$ and $P_{2i+1,t+1} \leftarrow P_{i,t} \cap A^{>}_{i,t}$.

Let $f : \mathcal{X} \to [0, 1]$ be given by $f(x) = i/2^{T-1}$ for all $x \in P_{i,T-1}$ and all $i \in [2^t]$.
**return** $f$

---

set of all maximal elements of $h^{-1}_{i,t}([i/2^t, (i + 1/2)/2^t])$. Under this notation, we have that

$$\text{score}_{i,t}(A_{i,t}) - \min_{A \in \mathcal{A}_{i,t}} \text{score}_{i,t}(A)$$

$$\geq \text{score}_{i,t}(A_{i,t}) - \text{score}_{i,t}(\tilde{A}_{i,t}) \tag{5}$$

$$= \left( \mathcal{L}^{\text{abs}}_{[i/2^t, (i+1)/2^t]}(h_{2i,t+1}; D_{2i,t+1}) + \mathcal{L}^{\text{abs}}_{[i/2^t, (i+1)/2^t]}(h_{2i+1,t+1}; D_{2i+1,t+1}) \right)$$
$$- \mathcal{L}^{\text{abs}}_{[i/2^t, (i+1)/2^t]}(h_{i,t}; D_{i,t})$$

$$= \mathcal{L}^{\text{abs}}(h_{2i,t+1}; D_{2i,t+1}) + \mathcal{L}^{\text{abs}}(h_{2i+1,t+1}; D_{2i+1,t+1}) - \mathcal{L}^{\text{abs}}(h_{i,t}; D_{i,t}). \tag{6}$$

Finally, notice that

$$\mathcal{L}^{\text{abs}}(f; D_{i,T-1}) - \mathcal{L}^{\text{abs}}(h_{i,T-1}; D_{i,T-1}) \leq \frac{L}{2^{T-1}} \cdot |D_{i,T-1}| = O\left(\frac{|D_{i,T-1}|}{\varepsilon n}\right). \tag{7}$$

With all the ingredients ready, we may now bound the expected (unnormalized) excess risk. We have that

$$\mathcal{L}^{\text{abs}}(f; D) = \sum_{i \in [2^{T-1}]} \mathcal{L}^{\text{abs}}(f; D_{i,T-1})$$

$$\overset{(7)}{\leq} \sum_{i \in [2^{T-1}]} \left( O\left(\frac{|D_{i,T-1}|}{\varepsilon n}\right) + \mathcal{L}^{\text{abs}}(h_{i,T-1}; D_{i,T-1}) \right)$$

$$= O(1/\varepsilon) + \sum_{i \in [2^{T-1}]} \mathcal{L}^{\text{abs}}(h_{i,T-1}; D_{i,T-1})$$

$$= O(1/\varepsilon) + \mathcal{L}^{\text{abs}}(h_{0,0}; D_{0,0})$$

$$+ \sum_{t \in [T-1]} \sum_{i \in [2^{t-1}]} \left( \mathcal{L}^{\text{abs}}(h_{2i,t}; D_{2i,t}) + \mathcal{L}^{\text{abs}}(h_{2i+1,t}; D_{2i+1,t}) - \mathcal{L}^{\text{abs}}(h_{i,t-1}; D_{i,t-1}) \right).$$

Taking the expectation on both sides and using (4) and (6) yields

$$\mathbb{E}[\mathcal{L}^{\mathrm{abs}}(f; D)] \leq O(1/\varepsilon) \, + \, \mathcal{L}^{\mathrm{abs}}(h_{0,0}; D_{0,0}) \, + \, \sum_{t \in [T-1]} \sum_{i \in [2^{t-1}]} O\left(\frac{L \cdot \mathrm{width}(\mathcal{X}) \cdot \log |\mathcal{X}|}{\varepsilon' \cdot 2^t}\right)$$

$$= O(1/\varepsilon) \, + \, \mathcal{L}^{\mathrm{abs}}(f^*; D) \, + \, \sum_{t \in [T-1]} O\left(\frac{L \cdot \mathrm{width}(\mathcal{X}) \cdot \log |\mathcal{X}|}{\varepsilon'}\right)$$

$$= O(1/\varepsilon) \, + \, \mathcal{L}^{\mathrm{abs}}(f^*; D) \, + \, O\left(T \cdot \frac{L \cdot \mathrm{width}(\mathcal{X}) \cdot \log |\mathcal{X}|}{\varepsilon'}\right)$$

$$= O(1/\varepsilon) \, + \, \mathcal{L}^{\mathrm{abs}}(f^*; D) \, + \, O\left(T^2 \cdot \frac{L \cdot \mathrm{width}(\mathcal{X}) \cdot \log |\mathcal{X}|}{\varepsilon}\right)$$

$$= \mathcal{L}^{\mathrm{abs}}(f^*; D) \, + \, O\left(\frac{L \cdot \mathrm{width}(\mathcal{X}) \cdot \log |\mathcal{X}| \cdot (1 + \log^2(\varepsilon n))}{\varepsilon}\right).$$

Dividing both sides by $n$ yields the desired claim. □