# OpenReview forum: "Private Isotonic Regression"
_NeurIPS.cc/2022/Conference — NeurIPS 2022 Accept_

### Official Review · Reviewer_YkU9 · 2022-07-10

**Rating:** 7
**Confidence:** 4
**Soundness:** 4 excellent
**Presentation:** 4 excellent
**Contribution:** 4 excellent

**Summary:**

This paper is the first to deal with DP isotonic regression: where the domain is some *partially* ordered set X and the goal is to find a *monotonic* f:X\to[0,1] that minimizes a certain empirical loss. The paper first discusses the totally-ordered set X case and then the partially ordered set case by implementing a generalization of the totally-ordered algorithm.

This is a the first paper to tackle this problem.

**Questions:**

Many typos throughout (like "the the" on P.1). Please go over the paper carefully and remove them.

**Limitations:**

First paper to tackle a new problem, and as such the painting isn't "complete" yet.

**Strengths And Weaknesses:**

Strengths:
* first to deal with this problem
* poses upper- and lower-bounds that depend on both log(|X|) and width(X) (the max-length of an anti-chain in X).

Weakness:
* upper and lower bounds don't match yet fully
That is of course, expected from a first paper.

I think this is an interesting paper that is likely to instigate follow-up works on this version of ERM and many other variants of "constrained" ERMs. A clear accept.

---

> ### Author Response · Authors · 2022-08-02
> **Typos.**
>
> We apologize for the typos in the paper and thank you for bringing this to our attention.  We will make a more careful pass for the revision.

---

> > ### Comment · Reviewer_YkU9 · 2022-08-07
> > **No further questions**
> >
> > Also, no need to apologize for typos. They happen to us all, and all the tyme.

---

### Official Review · Reviewer_4CVD · 2022-07-11

**Rating:** 7
**Confidence:** 3
**Soundness:** 3 good
**Presentation:** 4 excellent
**Contribution:** 3 good

**Summary:**

The paper studies the problem of diferentially private isotonic regression. It first introduces an algorithm and its excess risk. It then studies a lower bound for solving this problem privately, and shows that the gap between the two bounds is tight, in the sense that for each bound there exist posets for which each bound is tight,  and thus the gap cannot be closed. The algorithm runs in near linear time for totally ordered sets with $\ell_1$ and $\ell^2_2$ losses.

Privacy is guaranteed by relying on the exponential mechanism to iteratively select threshold functions on smaller partitions of the domain. Then it applies standard composition to allocate the privacy budget across iterations.


**Questions:**

- Do these complexity bounds allow for running the algorithm in practice? or what scale of problems, settings would this algorithm fit and be meaningful?
- Is it useful to consider algorithms similar to the one in Kaplan 2020 [27] that could reduce the dependency on the domain size?


**Limitations:**

- Width(X) can be an extremely large quantity and make this algorithm impractical.


**Strengths And Weaknesses:**


*Strengths*
- The specific problem of isotonic regression has not been studied before in the privacy literature.
- The paper provides a clear characterization of the problem introducing upper and lower bounds, and assumptions that allow for improvement or tightness of the results.
- The paper is clearly written and well organized.

*Weaknesses*:
- The paper could provide more tangible intuition on the results, for example in what settings this would be a meaningful practical algorithm, and in what settings it is still a first attempt that needs improvements to be applied. Either a discussion or small synthetic experiments could help understand these results, especially given that there is no previous work on the area.  This would give an intuition on the price of privacy, the easiness to tune clipping, etc.

---

> ### Author Response · Authors · 2022-08-02
> **Scalability & Algorithm of [27].**
>
> * The running time of our algorithm is quasi-linear for the most commonly used $\ell_1$ and $\ell_2^2$ losses (Lemma 13).  Hence it is scalable.
>
> * Thank you for the pointer.  The algorithm of [27] is for approximate-DP whereas our work is for pure-DP.  We will give a pointer to this reference and suggest this line of research as an interesting future direction.

---

> > ### Comment · Reviewer_4CVD · 2022-08-09
> > **Thanks for the response**
> >
> > Thanks for the clear response, I don't have any further comments.

---

### Official Review · Reviewer_heQ2 · 2022-07-11

**Rating:** 7
**Confidence:** 4
**Soundness:** 3 good
**Presentation:** 4 excellent
**Contribution:** 4 excellent

**Summary:**

This paper considers the problem of private isotonic regression in which given a dataset D consisting of n samples, the goal is to output a monotone function f that minimizes the empirical risk $L(f;D)=(1/n)\sum_i \ell(f(x_i,y_i)$. The authors give a pure DP algorithm for the most general version of the problem which considers a poset X and a Lipschitz loss function $\ell$ and they obtain an expected excess empirical risk of $width(X)*log(X)/n$.  For a fully ordered set, this algorithm is efficient and the idea is to privately choose a maximal point \alpha via the exponential mechanism and then recursively obtain the final function f by gluing together the functions obtained from recursing on the two partitions of [m] created via \alpha. The authors note that a simple implementation of assigning the (unnormalized) empirical risk as the score function results in a large error loss, so instead, they use a clipped version of the loss function as the score function resulting in reasonably low sensitivity. The more general DP algorithm has a similar flavor, except now one has to privately choose multiple maximal points \alpha, which leads to a less efficient algorithm (due to multiple calls to the exponential mechanism).
The authors also obtain a near-matching lower bound of $(width(X)+log(X))/n$. They achieve this by reducing to a known DP lower bound for DP algorithms that output a binary vector that is close to the input. They also show that while there is a gap between the demonstrated upper and lower bounds, there are posets that tightly realize each bound.


**Questions:**

See in comments above.

**Limitations:**

See in comments above. No potential for negative societal impact.

**Strengths And Weaknesses:**

Originality:
The contributions are original and would be of much interest to the overall machine learning community. It would be good to discuss existing DP algorithms on the closely related topic of simple linear regression in the Related Work section (see Daniel Alabi, Audra McMillan, Jayshree Sarathy, Adam D. Smith, Salil P. Vadhan: Differentially Private Simple Linear Regression. Proc. Priv. Enhancing Technol. 2022(2): 184-204 (2022))


Quality: The methods used are standard techniques in DP such as exponential mechanism, composition for the upper bounds, and reductions from known DP problems for the lower bounds. I am fairly confident that the work is sound, although I have not checked every single detail.


Clarity: The paper is mostly well-written, just for completeness, it would be good to explicitly state the running time of the DP algorithm for the general posets.


Significance: Isotonic regression is an important primitive in the machine learning toolbox. This work advances the state of the art on DP machine learning algorithms by adding this problem to the DP machine learning toolbox. The algorithms and proofs presented are relatively straightforward and easy to follow, and the authors also leave a set of intriguing open questions which may lead to further understanding of the complexity of machine learning tasks under DP constraints.

---

> ### Author Response · Authors · 2022-08-02
> **Thanks for the reference.**
>
> Thanks for the reference.  We will add this to the revision.

---

### Official Review · Reviewer_DQZ5 · 2022-07-18

**Rating:** 7
**Confidence:** 4
**Soundness:** 3 good
**Presentation:** 3 good
**Contribution:** 3 good

**Summary:**

This paper considers the problem of DP isotonic regression. For general loss functions, an inefficient algorithm is proposed to achieve a utility with a logarithmic dependency on the alphabet size for pure dp, An efficient algorithm is provided for $\ell_1$ and $\ell_2$ loss functions.

**Questions:**

NA

**Ethics Review Area:**

["I don’t know"]

**Strengths And Weaknesses:**

Strengths:
1. This paper considers the problem of DP isotonic regression. For general loss functions, an inefficient algorithm is proposed to achieve a utility with a logarithmic dependency on the alphabet size for pure dp, The result is tight if the paper considers the minimax setting. I think this is a very good result for an initial work in this area.
2. An efficient algorithm is provided for $\ell_1$ and $\ell_2$ loss functions.

Weakness:
1. My most complaints are on the presentation. First, the paper does not distinguish between the minimax setting and the instance based setting. Specifically, I was quite confused when I first looked at the discussion about the tightness. If the paper explicitly defined the two concepts, it could easily say the algorithm is minimax optimal but not instance optimal. Furthermore, it would be much better if the authors could define the isotonic regression and the width in the introduction, making it easier for the readers to evaluate the results.
1. it remains interesting to see the effects of different loss functions, both to the sample complexity and computation complexity.

Minor:
1. No hyperlinks for theorems.

---

> ### Author Response · Authors · 2022-08-02
> **Clarification**
>
> * We do not study instance optimality in this work - we will clarify this in the revision.  We will define isotonic regression and width in the introduction.
>
> * Thanks for the comments on different loss functions.  We will add this to our list of future research directions.
>
> * Thanks for pointing out - we will add hyperlinks to the Theorems

---

> > ### Comment · Reviewer_DQZ5 · 2022-08-08
> > **Thank you for the response**
> >
> > Thank you for the response, which has resolved my questions.

---

### Meta-Review · Area_Chair_MxDU · 2022-08-23

**Recommendation:** Accept
**Confidence:** Certain

**Metareview:**

Most reviewers found the paper well written with no serious doubts regarding the correctness. We hope authors incorporate the comments from the reviewers in their final revision to improve the presentation.

**Award:**

No

---

### Decision · Program_Chairs · 2022-09-14

Accept